# Osteocytes Influence on Bone Matrix Integrity Affects Biomechanical Competence at Bone-Implant Interface of Bioactive-Coated Titanium Implants in Rat Tibiae

**DOI:** 10.3390/ijms23010374

**Published:** 2021-12-29

**Authors:** Sabine Stoetzel, Deeksha Malhan, Ute Wild, Christian Helbing, Fathi Hassan, Sameh Attia, Klaus D. Jandt, Christian Heiss, Thaqif El Khassawna

**Affiliations:** 1Experimental Trauma Surgery, Faculty of Medicine, Justus-Liebig-University, Aulweg 128, 35392 Giessen, Germany; Sabine.stoetzel@chiru.med.uni-giessen.de (S.S.); Deeksha.Malhan@chiru.med.uni-giessen.de (D.M.); Ute.wild@med.uni-giessen.de (U.W.); Fathihassan@hotmail.com (F.H.); christian.heiss@chiru.med.uni-giessen.de (C.H.); 2Chair of Materials Science (CMS), Otto Schott Institute of Materials Research (OSIM), Faculty of Physics and Astronomy, Friedrich Schiller University Jena, Löbdergraben 32, 07743 Jena, Germany; christian.helbing@gmx.de (C.H.); k.jandt@uni-jena.de (K.D.J.); 3Department of Oral and Maxillofacial Surgery, Justus-Liebig University of Giessen, Klinikstrasse 33, 35392 Giessen, Germany; Sameh.Attia@dentist.med.uni-giessen.de; 4Department of Trauma, Hand and Reconstructive Surgery, Justus-Liebig University Giessen, Rudolf-Buchheim-Strasse 7, 35392 Giessen, Germany

**Keywords:** titanium implants, gelatin, hyaluronic acid, osseointegration, osteocytes, bioactive coating

## Abstract

Osseointegration is a prerequisite for the long-term success of implants. Titanium implants are preferred for their biocompatibility and mechanical properties. Nonetheless, the need for early and immediate loading requires enhancing these properties by adding bioactive coatings. In this preclinical study, extracellular matrix properties and cellular balance at the implant/bone interface was examined. Polyelectrolyte multilayers of chitosan and gelatin or with chitosan and Hyaluronic acid fabricated on titanium alloy using a layer-by-layer self-assembly process were compared with native titanium alloy. The study aimed to histologically evaluate bone parameters that correlate to the biomechanical anchorage enhancement resulted from bioactive coatings of titanium implants in a rat animal model. Superior collagen fiber arrangements and an increased number of active osteocytes reflected a significant improvement of bone matrix quality at the bone interface of the chitosan/gelatin-coated titan implants over chitosan/hyaluronic acid-coated and native implants. Furthermore, the numbers and localization of osteoblasts and osteoclasts in the reparative and remodeling phases suggested a better cellular balance in the chitosan/Gel-coated group over the other two groups. Investigating the micro-mechanical properties of bone tissue at the interface can elucidate detailed discrepancies between different promising bioactive coatings of titanium alloys to maximize their benefit in future medical applications.

## 1. Introduction

Implants are widely used in orthopedics and dentistry such as joint arthroplasty [1], hip replacement [2,3], maxillofacial reconstruction [4,5,6], especially intra oral implantation [7,8]. Besides biocompatibility and osteointegration, other aspects are important from the surgical view such as implant size, shape, material, and mechanical stability.

Therefore, metallic implants especially titanium (Ti)-based ones are the most common in both orthopedics and dentistry. The biomechanical performance [9], and bioinert nature of the Ti-alloy are among the reasons why Ti is widely used in metallic implants. Nonetheless, different bioactive coatings of Ti surface are used to enhance osseointegration and cell proliferation [10,11].

The choice of bioactive coating depends on the desired functional and biochemical properties. Biomolecules such as gelatin (Gel), and hyaluronic acid (HA) were successfully used as coating materials on titanium surfaces to enhance osseointegration using the layer-by-layer coating method [12,13,14,15,16,17,18]. Moreover, in vitro testing of Gel, and HA on human osteoblast cell lines showed enhanced cell proliferation compared with uncoated titanium implants [19]. Furthermore, preclinical testing of Gel and HA showed increased bone formation [20,21].

Cai et al. examined the effect of implant surface coating with chitosan (Chi)/Gel and Chi/HA on Ti-alloy using the layer-by-layer technique [22]. The coating was also examined by Zankovych et al. in rat tibia at two time points of healing, 3 weeks and 8 weeks [23]. The latter study reported that higher shear strength was needed to pullout bioactive-coated implants compared with native Ti-alloy. However, the cellular and bone matrix characteristics which might have led to this improvement were not fully investigated. The role of osteoblasts and osteoclast in bone formation and remodeling is crucial to investigate to examine new bone formation and remodeling. Furthermore, the quality of the extracellular matrix is crucial in healing and osteointegration. The maturity and quality of the bone matrix are reflected by osteocytes on the cellular level, and the arrangement and homogeneity of collagen fibers on the structural level. Two clinical studies have indicated a difference in osteocytes density around titan dental implants after at least 8 weeks [24] and up to 27 years of follow-up [25]. Therefore, this preclinical follow-up study aims to explore the differences in bone matrix mineralization and bone quality between Chi/Gel-coated, Chi/HA-coated, compared with uncoated Ti implants after 3 weeks and 8 weeks of intramedullary pin insertion in rat tibia using a detailed histological analysis. The study focuses on osteocytes and collagen fibers arrangement as the main indicators of bone matrix quality; furthermore, evaluating cellular changes within the bone-implant interface to gain insight to biomaterial success.

The previous study reported a higher biomechanical competence resulted from Chi/Gel and Chi/HA coatings [23]. This study hypothesized that Chi/Gel and Chi/HA coatings correlate with cellular changes in the bone matrix and osteocytes vitality, in accordance with the previously reported biomechanical testing. Indeed, the results show a significantly improved bone matrix quality, and increased osteocyte numbers within bone/implant interface in coated implants compared with uncoated controls. Whereas the interface around the Chi/Gel-coated titan implant reflects overall significantly enhanced extracellular matrix properties, and an increased number of active osteocytes. Furthermore, the qualitative and quantitative analysis of anabolic and catabolic cellular balance suggests a notable cellular balance in the Chi/Gel-coated group over the Chi/HA and Ti groups. The results of the current study conferred the importance of osteocytes and the surrounding extracellular matrix as indicative of biomechanical competence in the microenvironment of the bone/implant interface. This encourages investigating the micro-mechanical properties of the tissue as a determinant of the implant performance. Especially when immediate or early loading is part of the intended use of the developed implant as a medical device.

## 2. Results

### 2.1. Bioactive Coating Accelerated Bone Matrix Mineralization within Bone-Implant Interface in All Groups after 8 W

Enhanced bone matrix mineralization within the bone-implant interface is essential for implant success. Therefore, the modified Masson Goldner stain was carried out to measure the mineralized bone and non-mineralized bone matrix area within the bone-implant interface (Figure 1). The mineralized bone matrix was seen as blue and the non-mineralized bone matrix was seen as orange using modified Masson Goldner stain (Figure 1A).

Although the mineralized bone matrix portion was higher in all groups with healing progress (8 W compared with 3 W, Figure 1B). The Chi/Gel group showed a significantly higher mineralized bone matrix area after 8 W compared with 3 W (*p* = 0.008). Ti and Chi/HA also showed a higher mineralized bone matrix area after 8 W compared with 3 W (*p* = 0.03 and *p* = 0.01, respectively). At 3 W, a higher portion of mineralized bone matrix were seen in the Chi/HA group compared with the Ti and Chi/Gel groups (Figure 1B). Whereas the Ti group showed the lowest mineralized bone matrix area at 3 W. After 8 W, the Chi/Gel group showed a higher portion of mineralized bone when compared with the Ti and Chi/HA groups. Whereas the Ti group showed the lowest portion of the mineralized matrix at 8 W (Figure 1B). No other significant differences were observed between the groups within the same time point.

All groups showed a lower area of non-mineralized bone matrix after 8 W compared with 3 W (Figure 1C). Chi/Gel showed a significantly lower non-mineralized bone matrix area after 8 W compared with 3 W (*p* = 0.006). After 3 W, the non-mineralized bone matrix portion was lower in the Chi/HA group compared with the Ti and Chi/Gel groups (Figure 1C). Whereas the Ti group showed the highest portion of non-mineralized bone matrix at 3 W. After 8 W, Chi/Gel showed the lowest portion of non-mineralized bone matrix compared with the Ti and Chi/HA groups (Figure 1C). The Ti group showed the highest portion of non-mineralized bone matrix at 8 W compared with other groups. No significant differences were observed.

The improved bone matrix quality after 8 W as seen through histomorphometry encouraged the further evaluation of bone matrix quality.

### 2.2. Higher Number of Osteoblasts and Osteoclasts within Bone-Implant Interface in Ti and Chi/Gel Groups after 8 W

Bone remodeling is balanced through well-regulated osteoblast and osteoclast activity. Therefore, the number of osteoclasts and osteoblasts were examined within the bone-implant interface through modified Masson Goldner stain.

A higher number of osteoblasts was seen within the Chi/HA group compared with Ti and Chi/Gel groups after 3 W. Whereas, a lower number of osteoblasts were seen within the Chi/Gel group after 3 W. Interestingly, a higher number of osteoblasts were seen within Ti group compared with Chi/Gel and Chi/HA groups after 8 W.

A higher number of osteoclasts were seen within Chi/HA group compared with Ti and Chi/Gel groups after 3 W. A lower number of osteoclasts were seen within the Chi/Gel group after 3 W (Figure 2). Whereas after 8 W, a higher number of osteoclasts was seen within Chi/Gel group compared with other groups (Figure 2) No significant differences were observed. The mean ± SEM values of quantitative analysis of osteoblasts and osteoclasts count are shown in Appendix A.

### 2.3. Higher Amount of Spindle-Shaped Osteocytes within Bone-Implant Interface in all Groups after 8 W

Osteocytes are amongst the main regulators of bone matrix integrity. Therefore, osteocytes morphological changes were investigated using silver nitrate stain. Morphologically, osteocytes are categorized as spindle-shaped (active), spherical shaped (intermediate), and inactive (empty lacunae) (Figure 3(A1–A3)). 

Total osteocytes count between time points did not change in the Ti group from 3 W to 8 W. However, lower total osteocytes number was seen with a trend in Chi/HA group (*p* = 0.06) and significantly in the Chi/Gel group (*p* = 0.028) at 8 W compared with 3 W (Figure 3B). The Chi/HA group showed the highest number of total osteocytes and the Ti group showed the lowest at 3 W and 8 W; however, no significant differences were observed. 

Spindle-shaped osteocytes were higher in all groups after 8 W compared with 3 W (Ti group *p* = 0.032, Chi/Gel group *p* = 0.01, Chi/HA group *p* = 0.055, Figure 3C). The Chi/Gel group showed the highest spindle-shaped osteocytes compared with Ti and Chi/HA after 3 W and 8 W. Whereas, the Chi/HA group showed the lowest number of spindle-shaped osteocytes compared with Ti and Chi/Gel after 3 W and 8 W. However, no significant differences were seen.

The intermediate osteocytes count was not significantly lower in all three groups after 8 W compared with 3 W. The Chi/Gel group showed the lowest amount of intermediate shaped osteocytes after 3 W and 8 W compared with Ti and Chi/HA groups. Whereas Chi/HA group showed the highest intermediate shaped osteocytes count after 3 W and 8 W compared with other groups (Appendix A).

The empty lacunae count was not significantly lower in all groups after 8 W compared with 3 W. After 3 W, the Chi/Gel group showed the lowest count of empty lacunae compared with Ti and Chi/HA groups after 3 W and 8 W. Whereas the Chi/HA group showed a higher empty lacunae count compared with both groups after 3 W and 8 W (Appendix A). 

The mean ± SEM values of the osteocytes count normalized to bone area are shown in Appendix A. The percentage distribution of spindle-shaped, intermediate, and empty lacunae after 3 W and 8 W are shown in Appendix A.

### 2.4. Positive Correlation between Spindle-Shaped Osteocytes Count and Implant Anchorage

Biomechanical examination using the pullout test showed significantly higher shear strength in Chi/Gel and Chi/HA groups at 8 W compared with the titanium alloy control group (*p* < 0.05). The detailed results were previously published [23]. However, the previous data were used to correlate the shear strength with the current histological findings. At 3 W, the total number of osteocytes showed only one significant positive correlation (correlation coefficient (CC)), 1.0 with a *p*-value ≤ 0 in the Chi/Gel group. At 8 W the CC in the Chi/Gel group was −0.500; however, no statistical significance was seen. 

The correlation to spindle osteocytes was negative at 3 W in the Ti group (CC = −738, *p* = 0.26), and positive in the Chi/Gel group (CC = 0.816, *p* = 0.18), while no correlation was seen in the Chi/HA group. A positive correlation was seen in all groups at 8 W, CC = 1.000 for all; however, the correlation was significant in Chi/Gel and Chi/HA groups.

The biomechanical testing showed a very fluctuating results in the Ti group at 3 W (mean ± SD), 0.34 ± 0.41, respectively, N/mm^2^ and at 8 W, 0.46 ± 0.34, respectively. While the Chi/Gel group showed an improved shear strength at from 3 W to 8 W 0.076 ± 0.042 and 0.81 ± 0.24 N/mm^2^, respectively. The Chi/HA group showed a lower shear strength at 3 W with 0.1 ± 0.028, respectively, which improved at 8 W with 0.82 ± 0.31 N/mm^2^, respectively.

### 2.5. Alteration in Collagen Fibers Arrangement Correlates to Progression of Tissue Maturation and Materials Coating

The arrangement of collagen fibers reflects the quality of extracellular matrix (ECM). Qualitative assessment in Sirius red stain, which shows discrepancies in type I, type II, and type III collagen areas under polarized light, showed no differences between the groups and the time point. However, the examining fibers orientation of type I collagen areas was quantified using the CT-FIRE program (Figure 4A–C). To identify the properties of collagen fibers embedded within the matrix at the bone material interface, the Siris Red images were processed to monochromic images to measure the fibers number, alignment, and angle, length, width, and straightness.

The number of fibers reflects the density of the extracellular matrix around the implant. Although the number of Type I collagen fibers was not affected by the coating, the distribution around the implant and between time points changed. Despite no statistically significant alteration in fiber numbers in any group, the pattern reflects a slightly lower number after 8 W in all groups compared with 3 W (Figure 4D).

Regularly and irregularly oriented collagen fibers were seen at both time points. However, fibers closer to the coated implants were longer and better aligned compared with uncoated titan implants. The fiber angle indicates the orientation of bone mineralization pattern. Fibers are considered isotropic in nature when no overall alignment is seen (scored with a zero), whereas perfectly arranged fibrils are considered purely anisotropic (scored with one).

Fiber alignment in the Ti group was increased from 3 W to 8 W. In the Chi/Gel group it slightly decreased, while in the Chi/HA it slightly increased. Nonetheless, no statistically significant changes were seen in any group between time points. However, at 3 W the Chi/Gel and Chi/HA-coated implants showed a higher alignment value compared with the uncoated Titan implant (*p* = 0.022 and *p* = 0.024 respectively, Figure 4E). The alignment correlates to the angle of orientation, no differences of fiber angle in the Ti group were seen with the progression of tissue maturation, the angle mean was below 50° for all groups at 3 W. While at 8 W the mean was lower than the Chi/Gel group with 101.25° and close to 50° in the Ti and the Chi/HA groups with 45° (*p* = 0.06) and 54° (*p* = 0.03), respectively. Throughout the progression of tissue maturation only the Chi/Gel group showed a significant difference *p* = 0.017 between 3 W and 8 W (Figure 4F).

Fiber length is an important parameter in the determination of biomechanical properties. The Ti group had longer fibers at 3 W than 8 W without statistical significance, the Chi/Gel group showed longer fibers at 3 W than 8 W (*p* = 0.02), while the Chi/HA groups showed longer fibers at 8 W than 3 W (*p* = 0.031). However, the groups showed only a statistically significant difference between Chi/Gel and Chi/HA at 3 W (0.045) and a trend (*p* = 0.057) at 8 W (Figure 5A). Fiber width reflects the collagen networking; however, no remarkable discrepancies were seen at any time point in any group (Figure 5B). Fiber straightness reflects the anisotropic nature of the matrix, which indicates its quality, a higher straightness value at 3 W than the at 8 W was seen in the Ti group (*p* = 0.038), and the Chi/Gel group (*p* = 0.045). However, no difference was seen between the groups except at 8 W was between the lower values of the Ti group compared with the Chi/HA group (*p* = 0.045).

## 3. Discussion

Titanium implants are majorly used in orthopedics and dentistry because of their biocompatibility and mechanical properties [26,27]. However, enhancing the osseointegrative properties of titanium alloy can reduce implant failure [12,28,29]. Furthermore, clinical practice encourages early and immediate loading of implants to reduce healing time and gain mobility sooner. In many cases healing periods without implants cause functional disturbances. Such as in the case of edentulous patients with removable dentures [30,31]. In orthopedic and trauma surgery early mobilization has proven to be beneficial for patients and positively influenced healing [32]. Therefore, a variety of surface coatings and surface modifications were reported [15,16,17,18], aiming to improve titanium implant osseointegration. Bioactive materials such as calcium phosphate [19], collagen [20], gelatin [19], chitosan [20], and HA [21] are among the most commonly used materials for surface coating.

Layer-by-layer coating of Chi/Gel and Chi/HA, on a titanium surface was reported to enhanced in vitro cell proliferation [33]. Moreover, in vivo testing of bioactive-coated implants in a rat model showed enhanced biomechanical competence [23]. Therefore, the purpose of the present study was to explore the enhanced biomechanical competence resulted from the bioactive coating and investigate cellular and extracellular discrepancies at the bone implant interface. Such a detailed analysis can reveal the difference between the two coatings which was not deduced by biomechanical testing.

The tissue formed around the implant depends on cellular activity to reach optimal ECM deposition that signifies a qualitative bone healing and leads to functional osteointegration of the implant. Therefore, implant coatings aim to enhance the mineralization of the newly formed matrix at the implant interface to increase functionality or reach the required stability earlier than uncoated implants. The study showed coated implants indeed have a larger mineralized bone area around the implant at 3 W which increased at 8 W. Whereas the mineralized bone area around the uncoated implant was lower and did not change after 5 weeks of healing. In contrast the non-mineralized bone area, which resamples the formation of a new tissue, was highest around the uncoated implant at both time points. This suggests that the formation of a new matrix is similar with and without coating; however, the mineralization which signifies the matrix maturation and better osteointegration is better with the coated implants. Although after 8 weeks the healing should have been fully reached, the inferior mineralization explains the lower shear strength in the Ti group at both time points compared with the other groups.

New bone formation and degradation of the old bone matrix are part of the healing and osteointegration processes. Osteoblasts initiate matrix mineralization via different factors, especially collagen family proteins, form, the most abundant in ECM is the type I collagen [34]. However, the newly formed tissue is woven bone and has lesser mechanical stability compared with mature bone [35]. Therefore, osteoclasts mediated bone remodeling is important to reshape bone into lamellar bone through a balanced osteoblasts and osteoclasts activity [36]. The bone remodeling phase resulted in complete mineralization of bone matrix with bony bridging in rats is expect at 8 weeks. The study shows a delay in osteoblasts and osteoclasts recruitment in the Ti group were the number of osteoblasts increased at 8 W. This suggests that the matrix building is delayed and therefore the maturation and biomechanical competence of the tissue. The Chi/Gel group reflected the expected and favored cellular behavior through higher osteoblasts than osteoclast at 3 W and higher osteoclasts than osteoblasts at 8 W. This suggests an enhanced bone formation at the early time point and bone remodeling at the later time point. The Chi/Ha group showed similar numbers in both cell types at both time points; however, higher at 3 W than at 8 W. This suggests that the coating enhanced both osteoblasts and osteoclasts, which showed a positive effect on the mineralization process. However, the quality of the matrix was investigated by the number of osteocytes especially spindle-shaped ones and the collagen fiber arrangements. The Ti group showed the lowest number of osteocytes at both time points. However, the spindle shape that indicates mature bone matrix with viable cells, and were highest in the Chi/Gel group. The data also showed that the Chi/HA group had higher number of osteocytes but lower spindle-shaped osteocytes than the Ti -group at both time points. This was supported through the enhanced properties of collagen fibers in the Ti compared with the Chi/HA with higher count and orientation angle, width, and alignment. The Chi/Gel group showed an enhanced matrix quality when investigating collagen fiber properties. Taken together the data suggest an enhanced quality in bone matrix in Chi/Gel-coated group, which supports the shear strength results of the pullout test.

The matured bone is composed of mechanically stable bone with a well-organized osteocytes network [37]. Osteocytes are the master regulator of bone mechanosensation and mechanotransduction [38]. Morphological changes in osteocytes and their count govern the bone quality. Besides osteocytes, collagen fibers are an integral part of ECM in providing mechanical support [39]. Therefore, the current study emphasizes particularly on osteocyte specific changes in all three groups within a bone-implant interface from the same animals utilized in the previous study [23] and the structural integrity and orientation of the collagen fibers.

The implant osseointegration is evaluated through a profound connection between bone and implant surface at the cellular level [40]. Osseointegration is established through bone remodeling, which involves a cross talk between bone forming osteoblasts and bone resorbing osteoclasts [41]. The enhanced osseointegration of the bioactive-coated Ti implants was evident through histological analysis and the improved matrix mineralization (Figure 1), which accords with the previously reported biomechanical results [23]. The larger portion of non-mineralized matrix after 8 W indicates inferior healing in the Ti group compared with the other groups.

The composition of biomaterial and their surface structure affects osteoblast [42,43] and osteoclast activity [44,45]. Higher osteoblast and osteoclast count in the Chi/HA group compared with the Chi/Gel group and Ti group after 3 W contributes to the higher osteoprogenitor activity because of HA. However, both osteoblast and osteoclast counts were lower in ChiI/HA group compared with other groups after 8 W. However, without statistical significance. This observation correlates with the previous report which showed higher gene expression of osteoblast and osteoclast markers in HA-coated implants after 3 W but a decline after 12 W in rabbits [46]. Intriguingly, higher counts of both osteoblasts and osteoclasts in Chi/Gel and Ti were seen at 8 W compared with 3 W. The higher count of osteoblasts in Chi/Gel at 3 W indicates cell recruitment for bone matrix formation and mineralization. However, the high deviation of cell counts in the Chi/Gel and the Ti groups at 8 W might explain the unexpected high osteoblasts count. While the higher count of osteoclast in Ti and Chi/Gel groups reflects the remodeling stage. Nonetheless, the presence of osteoclasts at 3 W can be explained by the effect of implant roughness. Previous reports showed an increase in osteoclast and osteoblast activity in vitro with a change in surface roughness of the Ti implant [47,48]. This implicates that the osteoblast and osteoclast count vary with bioactive coating and surface roughness and warrants further investigation into the roughness of the presented coating materials. Nonetheless, it is known that osteocytes regulate both osteoblast and osteoclast activity under mechanical stress [49].

Furthermore, at the cellular level the osteocytes network predicts the biomechanical potential of the implant. Few clinical studies reported the changes in osteocyte density [24,25] and ultrastructural osteocyte organization [50] around dental implants. preclinically, morphological changes within osteocytes due to mechanical load was studied using scanning electron microscopy (SEM) in rabbit tibiae [51]. However, quantitative investigation of morphologically different osteocytes using osteocyte specific stain around bioactive-coated implants using light microscopy is reported in this study for the first time. Furthermore, the quantification and examination of extracellular matrix quality beyond mineralization using an in-depth analysis of collagen fiber quantification and orientation is also presented at the bone/implant interface for the first time.

Osteocytes respond to mechanical stress. In this study, all three types of osteocytes spindle (active), intermediate (round), and empty lacunae [52] were seen within the bone-implant interface of Ti, Chi/Gel, and Chi/HA groups (Figure 3A and Appendix A). Osteocytes form after osteoblasts are embedded in the mineralized bone matrix [37]. Therefore, the higher number of total osteocytes in Chi/Gel and Chi/HA groups compared with Ti after 3 W and 8 W suggests a faster bone matrix maturation after new bone matrix formation. This correlates with previous reports where higher osteocyte density was observed using SEM in bioactive-coated implants compared with non-coated implants in rabbits [53]. The changes also accord with the bone matrix mineralization seen in the Mason Goldner stain. Intriguingly, the total number of osteocytes did not change from 3 W to 8 W in the Ti group. Nonetheless, individual evaluation of spindle-shaped osteocytes, intermediate osteocytes, and empty lacunae showed changes within all three groups. Interestingly, in all groups higher spindle-shaped osteocytes count were seen after 8 W compared with 3 W. The higher number of spindle-shaped osteocytes suggests an increased connectivity between active osteocytes and bone tissue. This observation also correlates with a previous report where an increased osteocyte-canaliculi network was observed using Transmission Electron Microscopy (TEM) within bone implant interface in human patients [54] and using SEM in rabbit tibiae [51].

The intermediate osteocytes and empty lacunae count were higher in all three groups at 3 W compared with 8 W. Osteocytes are unique cells, their apoptosis has been reported to initiate bone remodeling [55], which should start after 3 W. Nonetheless, intermediate osteocyte is a term used to describe round osteocytes, without distinguishing between apoptotic osteocytes turning from spindle to empty lacunae or new cells transforming from osteoblasts to spindle osteocytes. However, the lower count of both intermediate osteocytes and empty lacunae along with the increase in spindle-shaped osteocytes after 8 W indicates an improved mechanical stability and tissue maturation. Furthermore, the lower number of intermediate osteocytes and empty lacune, reflects on the reduction of the total number of osteocytes at 8 W. This aligns with previous studies which showed that the osteocytes density declines once newly formed bone reaches the mechanical competence of [25].

The higher spindle-shaped osteocytes in both Chi/Gel and Chi/HA-coated implants direct towards enhanced cell-interactive properties of the implant. Vlieberghe et al. reported enhanced cell-interactive properties because of Gel coating in vitro testing [56]. While HA-coating was reported to promote cellular interaction with implant surface [53]. Furthermore, the orientation angle of collagen fibers also reflects the quality of the bone matrix. At an early timepoint in all groups and at a later time point in the Ti and Chi/Ha group the mean angle was 45°, whereas in the Chi/Gel at 8 W the orientation angle mean was about 100°. Literature supports that bone quality transformation is associated with collagen fibers orientation angle and a misalignment angle of 50°. Therefore, interpreting bone as brittle at orientation angles less than 50° and as ductile above 50° [57]. Therefore, the collagen fiber orientation angle lower than 50° as in all groups and time points apart from Chi/Gel 8 W, reflect the isotropic structure of matrix around the implant. Therefore, the higher angle indicates higher bone stiffness [58]. Fiber alignment was lower at 3 W and showed large deviation in the Ti-group at 8 W, while in the coated implants a more homogenous alignment was seen with a mean of 0.5. This result supports a more isotropic nature of bone around the native titanium. The CT-FIRE program, scores the misalignment (no overall alignment) with a zero and consider them as isotropic. Whereas perfectly arranged fibrils are considered purely anisotropic and scored with one. Taken in consideration that the groups showed no significant difference in fiber counts at any time point, the data suggest that the bioactive coating influenced the orientation of the fibers and thusly the maturation of the matrix. The orientation hints to the biomechanical properties of the bone and not its mineralization. Nonetheless, the calcification of organic matrix adopts the fibers orientation discrepancies along the c-axis [59,60]. In other words, the fiber orientation differentiates the biomechanical properties between the matrices also when the mineralization degree is equal.

Fiber length also reflects the mechanical and structural properties of bone, all groups showed a length above the average fiber length of 1µm. Short fibers indicate the inferior mechanical stability of bone. Both tumor biopsies of patients [61] and bones of osteoporotic rat [62] showed high frequency of smaller collagen fibers. Fiber width was high at all time points and in all groups; however, it was highest at 8 W in the Chi/Gel and 3 W Ti. The results of fibers length and width are intriguing. Although they confirm the quality of bone in the Chi/Gel group, they also indicate good quality of the Chi/Ha and the Ti groups. However, this might be due to the relatively late time points of investigation in a healthy bone, which are at the edge of the reparative/remodeling phase at 3 W and at the end of the remodeling phase by 8 W [63,64]. Previous reports indicated that a compact and crosslinked collagen structure is formed after bone remodeling [65]. This might be also supported by the similar fiber straightness in all groups. Although coated implants show a similar fiber straightness value at 3 W and higher at 8 W, results from the uncoated implants cannot be considered severely inferior. However, fiber straightness indicates bone isotropic structure characteristics. The results show significantly higher straightness value and thusly a more anisotropic structure at 8 W in the Chi/Gel group. The more anisotropic the matrix structure is the better are the biomechanical properties [66,67].

Taken together, the data suggest enhancement in the extracellular matrix of both groups with coated titanium alloy compared with the native one. However, a notable osteocytes connectivity and better collagen fibers properties suggest an overall better bone quality around the Chi/Gel-coated implants which increased with tissue maturation.

Nonetheless, the study is limited by the absence of earlier time points which hinders the examination of the initial cellular response to the coating. The low number of animals was a hindrance for examining the structure under a transmission electron microscope or performing further immunohistochemical analysis. Although, these factors limit the study from achieving a comprehensive analysis, those same factors are governed by the ethical regulation on animal experiments under the 3R-pricniples and not the researchers.

## 4. Materials and Methods

### 4.1. Animal Experiment

This study was performed in accordance with institutional regulations and German animal protection laws. The animal experiment was approved by the ethical committee of local governmental institution (Regierungspräsidium Thuringia, permit number/application number G02-008/10).

Male Sprague Dawley rats (age = 3 months) were randomly divided into three groups: (1) titanium alloy implant without coating (Control group, Ti), (2) titanium implant coated with chitosan and Gelatin (Chi/Gel), and (3) titanium implant coated with chitosan and Hyaluronic acid (HA). The rat model was used because of its suitability to the examined time points, where reports showed bone healing in young rats within 4 weeks [68]. Therefore, the study can report material anchorage and bone quality during the healing process (3 W) and after the healing is completed (8 W). The choice of male rats was to avoid the well-documented influence of sex hormones especially during the menstrual cycle in female rats [69].

Surgery was performed under general anesthesia by weight-adopted intraperitoneal injection of 0.15 mg/kg Body Weight (BW) Meditomidin (Domitor^®^, Pfizer, Berlin, Germany), 2.0 mg/kg BW Midazolam (Dormicum^®^, Ratiopharm, Ulm, Germany) and 0.005 mg/kg BW Fentanyl (Fentanyl^®^, Jans-sen-Cilag, Neuss, Germany). 

Animals were prepared for surgery: both hind legs were shaved and disinfected with alcohol. Sterile conditions were adhered to throughout surgery using sterile drapes and sheets. Both hind legs were covered with a sterile incision drape (Raucodrape^®^, Lohmann and Rauscher, Rengsdorf, Germany). A medial incision to expose the knee joint in both hind limbs was made 5 mm longitudinally, and a pilot hole marked at the intercondylar eminence. A custom-fabricated instrument, tip size 0.9 mm in diameter and 10 mm in length, was gradually rotated to create a channel from the proximal tibia epiphysis into the medullary canal. The implants were inserted via this channel and positioned 2 mm beyond the articulating cartilage.

Soft tissue was irrigated with sterile saline with fascia and skin incisions closed in a single-knot technique (Vicryl 5/0 and Prolene 5/0, Ethicon, Norderstedt, Germany). Prophylactic IM antibiotic (Terramycin^®^, Pfizer GmbH, Berlin, Germany) and analgesics (Buprenovet^®^, Bayer, Leverkusen, Germany) were administered once during surgery. The implant position was monitored and confirmed by X-ray imaging. General anesthesia was reversed by 0.75 mg/kg BW Atipamezol (Anti-sedan^®^, Pfizer, Berlin, Germany), 0.2 mg/kg BW Flumazenil (Flumazenil-hameln^®^, InveraArznei-mittel GmbH, Freiburg, Germany) and 0.12 mg/kg BW Naloxon (Deltaselect GmbH, Dreieich, Germany) [23].

Animals were euthanized after (weeks (W)) 3 and at 8 weeks post-surgery using intraperitoneal injection of sodium pentobarbital overdose, (200 mg/kg, Narcoren, 16 g/100 mL, Garbsen, Germany). Conformation of death followed through bilateral thoracotomy.

### 4.2. Histological Examination

After euthanasia, tibiae were harvested and cleaned of the surrounding soft tissue. The samples were then immediately fixed in 4% formalin and later infiltrated with ethanol and Technovit 7200VLC (HeraeusKulzer, Wehrheim, Germany). The samples were then embedded in PolyMethylMethaAcrylate resin (PMMA).

After embedding, PMMA blocks were prepared into 15–25 µm thick grindings to perform the histological analysis. First, modified Masson Goldner Stain was used to differentiate the mineralized (blue) and nonmineralized bone matrix (orange) within the bone-implant interface. Briefly, grindings were wiped in a 1:1 Acetone/Ethanol mixture, then submersed in hematoxylin for 35 min. The grindings were then washed in distilled water before submersion in ponceau-fuchsin solution for 20 min, then rinsed in 1% acetic acid solution for 1 min, then dipped in phosphotungstic acid orange G solution for 20 min before rewashed in 1% acetic acid solution for 1 min. The modification step was in submersing the grindings in Aniline-blue-orange-acetic-acid for 8 min, then washing in water, dehydrating in alcohol and finally mounting. Accordingly, changes in tissue mineralization in the different time points were qualitatively and quantitatively assessed around the implant. Additionally, osteoclasts and osteoblasts numbers in the newly formed bone were measured based on morphology in Masson Goldner-stained grindings. Further, silver nitrate staining was used to visualize osteocytes as the main cells in the extracellular matrix and their canaliculi network. Grindings were first decalcified using ethylenediaminetetraacetic acid (EDTA) for 1 week. The staining was carried out as described previously [70]. However, Sirius red stain was used to address the quality of the mineralized matrix around the implant. The stain differentiates type I, type II, and type III collagen within a sample under polarized light [71]. The collagen types appear yellow-red, blue-light yellow, and green, respectively. The stain was used to examine qualitatively the collagen types and quantitatively examine the organization and heterogeneity of collagen fibers orientation during callus using CT-FIRE program (detailed below). Quantification of fibril properties such as fiber angle, width, length, and straightness were carried out [72].

The length and width of fibers are calculated as pixel values. Whereas the alignment describes the scale of fibers direction starting from 0 that represents inhomogeneous orientations angles to 1 which suggests that all fibers are oriented with the same angle. Straightness also describes a scale on which the distance between the fiber endpoint is divided by the distance along the path of the fiber. When the scale is at 0 suggests a fuddled fiber where at 1 it indicates a perfectly straight fiber [73].

### 4.3. Image Capturing and Histomorphometry

The analysis was carried out for the bone implant interface as the region of interest (ROI). Images were captured using a Leica microscope (DM5500 B with AF6000; Leica Microsystems GmbH, Wetzlar, Germany). Then, 20× magnified images were captured from both Masson Goldner and silver nitrate-stained grindings.

Bone parameters such as areas of mineralized bone, non-mineralized bone, and total bone were calculated using ImageJ based plugin as described before [37]. The Region of interest (ROI) encompassed 50 µm thick ring around the implant. The mineralization was assessed as an indication of bone quality after healing, the mineralized areas are expected to increase with the progression of the healing while non-mineralized matrix shall decrease. The empty areas which have neither mineralized nor non-mineralized tissue were also calculated but not shown (Appendix A).

The count of osteocytes, osteoblasts, and osteocytes was manually counted under 40× magnification.

Sirius red sections were visualized using Leica microscope equipped with an additional Polarized analyzer (11555079, Analyzer 180, Rotatable; Leica Microsystem Ltd., Wetzlar, Germany) and lambda filter (11513908, Lambda plate in a holder; Leica Microsystem Ltd., 15 Wetzlar, Germany). The analyzer was kept at 150° and lambda plate was kept at 75° for all samples. Monochrome and colored images were taken using 10×, 20×, 40×, and oil immersion (100×) magnification as previously described [74].

An automated standalone CT-FIRE (version 2.0 Beta) package from MATLAB (version R2017b, MathWorks, Natick, MA, USA) was used to computationally segment collagen fibers from Sirius red-stained monochrome pictures. Fiber properties such as fiber length, width, straightness, and angle were quantified. CT-FIRE was used before to quantify fiber structure taken from transmission electron microscopy or fluorescence microscopy [62].

### 4.4. Statistical Analysis

The statistical analysis was performed using the statistical package PASW 28.0 (IBM, SPSS Inc., Armonk, NY, USA). To compensate for the number of samples, analysis repetitions were performed. The obtained data were log transformed before statistical analysis, and showed a normal distribution in the histogram as well as in the QQ-plot. A two-tailed Spearman’s Rho was implemented to examine the correlation between the osteocytes and the sheer stress. As data were collected in fields of views in repeated measurements, the analysis was carried out with the help of the mixed model analysis, in which fixed and random effects are modeled. The background is that by counting different samples for the same individual, the measurements are not independent [75]. This relationship between the values is taken into account by specifying the corresponding covariance matrix. A cutoff of *p*-value ≤ 0.05 was set to indicate statistical significance. In other words, the number of animals varied between the timepoint and group is as follows: (Ti: *n* = 5 (3 W), *n* = 3 (8 W); Chi/Gel: *n* = 5/time point; Chi/HA: *n* = 4 (3 W), *n* = 5 (8 W). However, the bone/material interface was imaged in a different field of view between 15 and 25, which were evaluated. Three types of osteocytes were counted: spindle-shaped, round, and osteocytes with empty lacuna. These types can be considered as three different stages of the same variable. In this consideration, an evaluation as a mixed model can be considered, where the different Osteocyte types are considered as measurement repetition. Time points were considered as fixed effects, the random variables were the type of osteocytes.

## 5. Conclusions

This study provided an insight to morphological changes of osteocytes within the bone implant interface of Ti, Gel, and Chi/HA groups. The use of silver nitrate stain enabled the visualization of osteocytes and its canaliculi network under a light microscope. Moreover, the quantitative evaluation of osteoblast and osteoclast count within bone-implant interface helped in deciphering the changes related to bone remodeling. However, the deeper insight into material behavior at earlier time points would be very beneficial.

## Figures and Tables

**Figure 1 ijms-23-00374-f001:**
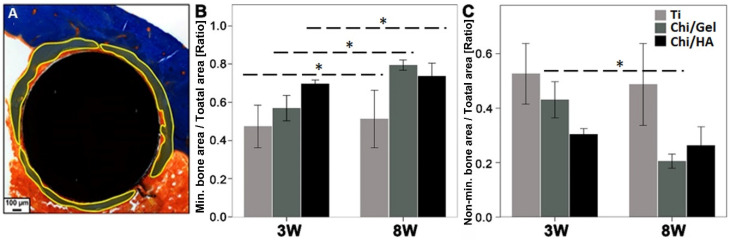
Higher amount of mineralized bone matrix seen within bone-implant interface after 8 W. Changes in matrix mineralization were evaluated using histological analysis. (**A**) Modified Masson Goldner stain helped in the quantification of mineralized and non-mineralized bone matrix area. (**B**) Chi/Gel group showed significantly higher mineralized bone matrix after 8 W compared with 3 W. Although not as apparent as the Chi/Gel group, Ti and Chi/HA groups also showed a higher mineralized bone matrix area after 8 W, (**C**) Chi/Gel showed a significantly lower portion of non-mineralized bone matrix area after 8 W compared with 3 W. Chi/HA group also showed lower non-mineralized bone area after 8 W (Ti: *n* = 5 (3 W), *n* = 3 (8 W); Chi/Gel: *n* = 5/time point; Chi/HA: *n* = 4 (3 W), *n* = 5 (8 W); *: *p* ≤ 0.05).

**Figure 2 ijms-23-00374-f002:**
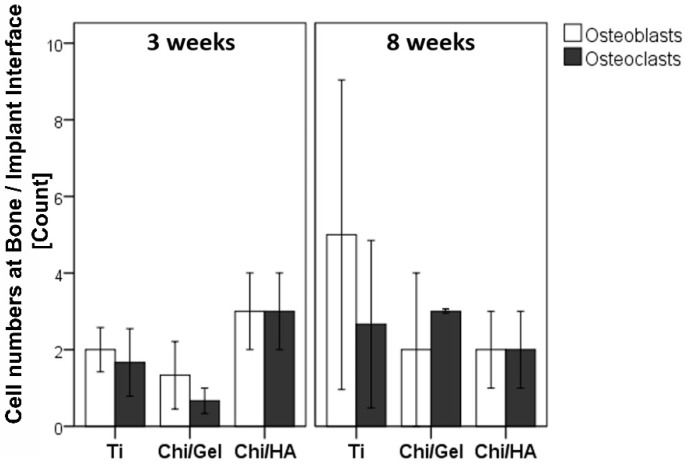
Quantitative evaluation of osteoblasts and osteoclasts within bone-implant interface showed higher osteoblasts/osteoclasts count in Ti and Chi/Gel groups after 8 W. Bone remodeling is governed by balanced osteoblast and osteoclast activity. Therefore, osteoblasts and osteoclasts were quantitatively evaluated for all groups. Higher osteoblasts and osteoclasts count were seen in Ti and Chi/Gel groups after 8 W compared with 3 W. Whereas, a lower number of osteoblasts and osteoclasts were seen in the Chi/HA group after 8 W compared with 3 W (Ti: *n* = 5 (3 W), *n* = 3 (8 W); Chi/Gel: *n* = 5/time point; Chi/HA: *n* = 4 (3 W), *n* = 5 (8 W).

**Figure 3 ijms-23-00374-f003:**
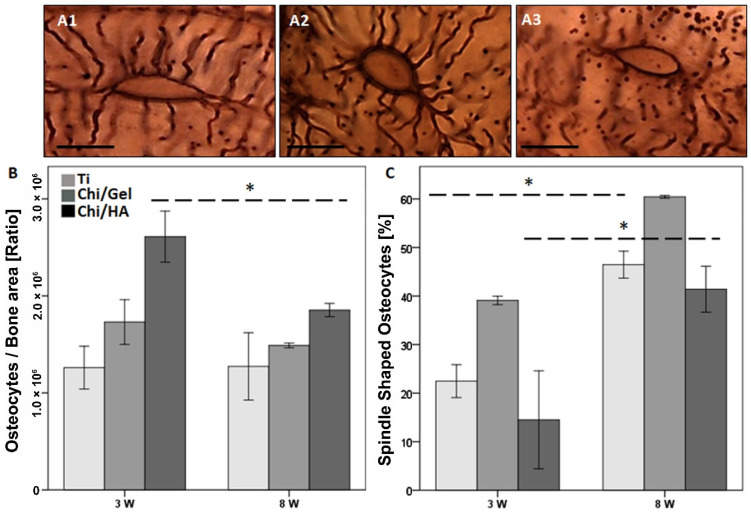
Quantitative evaluation of osteocytes within bone-implant interface showed higher spindle-shaped and lower empty lacunae in all groups after 8 W. Silver nitrate stain was used to visualize osteocytes within bone-implant interface. Morphologically, osteocytes are categorized as (**A1**) spindle-shaped, (**A2**) intermediate, and (**A3**) empty lacunae. (**B**) Total osteocytes were lower in Chi/Gel and Chi/HA groups after 8 W compared with 3 W. (**C**) Higher spindle-shaped osteocytes were seen in all groups after 8 W. Spindle-shaped osteocytes in the Chi/Gel group were higher compared with Ti and Chi/HA groups after 8 W. (Ti: *n* = 5 (3 W), *n* = 3 (8 W); Gel: *n* = 5/time point; HA: *n* = 4 (3 W), *n* = 5 (8 W); *: *p* ≤ 0.05, scale bar: 10 µm).

**Figure 4 ijms-23-00374-f004:**
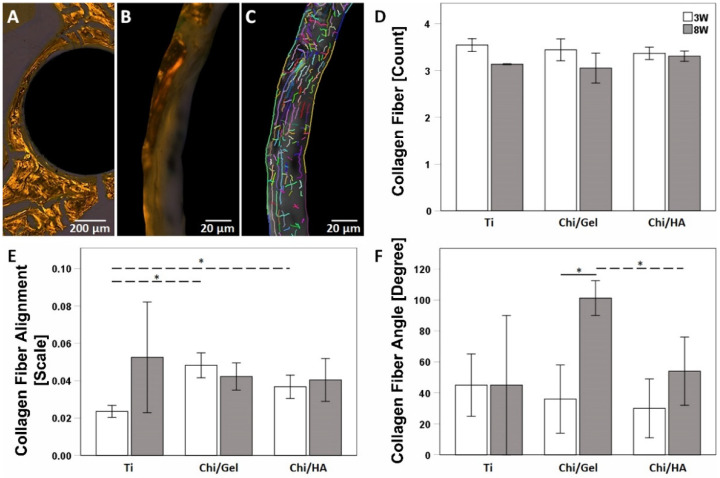
Collage fibers properties at the bone implant interface indicate bone matrix quality. (**A**) and (**B**) Sirius red-stained images were used to visualize collagen type around the implant. (**C**) collagen fibers properties were investigated on monochromic pictures using CT-FIRE software. (**D**) Number of collagen fibers was less at 8 W without significant difference to 3 W. (**E**) Fiber alignment was highest in the Chi/Gel group at 3 W, with progression of tissue maturation no significant change in fiber alignment was seen in any group. (**F**) Orientation angle of fibers was around or below 50°, at all time-points in all groups. (Ti: *n* = 5 (3 W), *n* = 3 (8 W); Chi/Gel: *n* = 5/time point; Chi/HA: *n* = 4 (3 W), *n* = 5 (8 W), **: p* ≤ 0.05).

**Figure 5 ijms-23-00374-f005:**
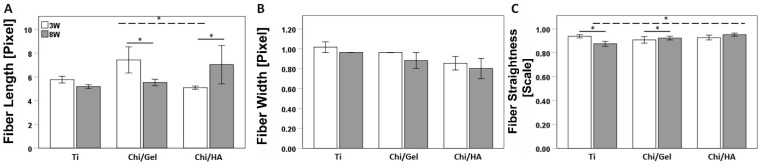
Enhanced collagen fiber properties led to enhanced micro-mechanical quality in Chi/Gel at 3 W and 8 W. CT-FIRE was used to quantify the collagen fibers from Sirius red-stained images. Collagen fibers orient mostly at smaller angles, <55°. (**A**) The fibers were mostly between 3 and 5 µm in length, the Chi/Gel group showed longer fibers at the early time point and shorter at the later time point, the Chi/Ha showed an inverse pattern. (**B**) Fiber width reflects crosslinking and intramolecular spacing, all groups showed comparable width within the range of healthy tissue. (**C**) Fiber straightness indicates anisotropic nature of the matrix, at 8 W the Chi/Gel and Chi/HA show the highest values. (Ti: *n* = 5 (3 W), *n* = 3 (8 W); Gel: *n* = 5/time point; HA: *n* = 4 (3 W), *n* = 5 (8 W), **: p* ≤ 0.05).

## Data Availability

Not applicable.

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
