# Peer review of "Osteocytes Influence on Bone Matrix Integrity Affects Biomechanical Competence at Bone-Implant Interface of Bioactive-Coated Titanium Implants in Rat Tibiae"

_ijms, 2021, doi:10.3390/ijms23010374_

Round 1
Reviewer 1 Report
The reviewed manuscript has practical relevance, yielding interesting applications, worth publishing, once the authors address some aspects, which in my opinion require their attention.
Although the article is overall well-argued, there are some parts of the manuscript which require attention, both as to its structure and English. I mention some indicative linguistic errors (only for page 1):
- Page 1, Line 39, “Medical implants…” I think the term “Medical” is redundant here, as all implants are of medical nature.
- Page 1, Line 40, dental implants fall within the general category of maxillofacial reconstruction. Please rephrase.
- Page 2, Line 46, “…make Ti more desirable.” This seems like an open statement, more desirable than what?
- Page 2, Line 49-50, “…cellular properties”. I think the term “cellular” is inappropriate. Bioactive material exhibit the capacity to interact with cells, these are not “cellular” properties but rather physicochemical or biochemical.
- Page 2, Line 54, “…and HA in showed…” I believe “in” is redundant here.
There are more of these syntax/linguistic issues throughout the text and I’d ask the authors to carefully proofread their manuscript again or consult a colleague versed in scientific English editing.
Abbreviations should be explained on first appearance (e.g. Page 2 line 56, “Chi”)
Also, keywords are used to assist search engines to find the article, as such, is seems redundant to use keywords apparent in the title.
What is the intention of figure 1 C? Mineralized and Non-mineralized bone matrix should sum-up to 1, except if there was an area around the implant that was not in contact with bone matrix at all (like in the case of uncoated Ti at the 3W timepoint, where figure 1b and 1C sum up to less than 100%). This is however difficult to evaluate when looking at two separate diagrams. It would make far morse sense to show this information in one bar diagram (e.g. will all bars lining up to 1 and portions of each bar showing mineralized/non-mineralized bone in different colors).
The variation in the Chi/Gel group of figure 2 (8W) is missing. It would help to see the lower values of those in white when superimposing black bars.
The discussion section is somewhat confusing. I would urege the authors to firstly explain the rationale behind their methodological approach, e.g. why is the quantity of osteoblasts and osteoclasts important and how does this and osteocyte shape correlate to bone mineralization. Once the own results have been discussed, then it would make sense to evaluate them with respect to literature.
Discussion should also include justification as to some methodological choices, e.g. why where only male rats used?
Methods should also describe the euthanasia process and how the Masson Goldner Stain was modified.
I’m not sure what the authors mean with the term “Biomechanical Competence” (used in the title) and how this relates to the presented results.
Reviewer 2 Report
Just some concerns which will need to be addressed:
Please give more detailed description of protocol of animal experiment – the statement that “it was described previously” is not acceptable. This manuscript present the results of completely different preclinical experiment on model animal and all essential information must be clearly given.
Statistics: The statistical description is not clear: “ to compensate for the number of samples, analysis repetitions were performed”. What was the number of observation for each trait/parameter per sample ? Please also name fixed and random effects in applied model. Was an individual considered as random effect ?
L71-84 This whole paragraph must be corrected. Working hypothesis of the present study should result from results of previous experiments, not vice versa (L73). Also no correlation was assessed in this study (L72). Most of the sentences in this section should be moved to the last paragraph of the discussion.
Figure 1. Not clear - What is the difference between mineralized and non-mineralized bone ? Do they sum up to 1 ? If yes, what is the idea of presenting both figures?
Figure 2. “Quantitative evaluation … showed higher osteoblasts/osteoclasts count … in groups after 8W” – it was not assessed with statistical analysis or the results are missing. The axis labelled as “cell number on implant interface” shows the total number of cell on the whole implant or per unit of length of the implant ?
Figure 3B – the same remark as previously. “Osteocytes /bone area” – means 1 osteocyte per E-6 of squared meter of bone ?
Figure 4F Angel ?
Figure 5 – units of physical quantities presented on figures are missing (length – m or um ?; width – ? )“straightness” – this trait was not defined in material and method section.
L46 Please provide a more recent reference than [9]
Round 2
Reviewer 2 Report
Thanks to the Authors for their diligent attention to the comments made.